# Software-Assisted Pattern Recognition of Persistent Organic Pollutants in Contaminated Human and Animal Food

**DOI:** 10.3390/molecules26030685

**Published:** 2021-01-28

**Authors:** Wenjing Guo, Jeffrey Archer, Morgan Moore, Sina Shojaee, Wen Zou, Weigong Ge, Linda Benjamin, Anthony Adeuya, Russell Fairchild, Huixiao Hong

**Affiliations:** 1National Center for Toxicological Research, U.S. Food & Drug Administration, 3900 NCTR Road, Jefferson, AR 72079, USA; Wenjing.guo@fda.hhs.gov (W.G.); Wen.Zou@fda.hhs.gov (W.Z.); Weigong.Ge@fda.hhs.gov (W.G.); 2Office of Regulatory Affairs, Office of Regulatory Science, Arkansas Laboratory, U.S. Food & Drug Administration, 3900 NCTR Road, Jefferson, AR 72079, USA; Jeffrey.Archer@fda.hhs.gov (J.A.); Morgan.Moore@fda.hhs.gov (M.M.); Sina.Shojaee@fda.hhs.gov (S.S.); Russell.Fairchild@fda.hhs.gov (R.F.); 3Center for Veterinary Medicine, U.S. Food & Drug Administration, 7500 Standish Place, Rockville, MD 20855, USA; Linda.Benjamin@fda.hhs.gov; 4Center for Food Safety and Applied Nutrition, U.S. Food & Drug Administration, 5001 Campus Dr, College Park, MD 20740, USA; Anthony.Adeuya@fda.hhs.gov

**Keywords:** persistent organic pollutant, software, similarity, congener pattern, contamination

## Abstract

Persistent Organic Pollutants (POPs) are a serious food safety concern due to their persistence and toxic effects. To promote food safety and protect human health, it is important to understand the sources of POPs and how to minimize human exposure to these contaminants. The POPs Program within the U.S. Food and Drug Administration (FDA), manually evaluates congener patterns of POPs-contaminated samples and sometimes compares the finding to other previously analyzed samples with similar patterns. This manual comparison is time consuming and solely depends on human expertise. To improve the efficiency of this evaluation, we developed software to assist in identifying potential sources of POPs contamination by detecting similarities between the congener patterns of a contaminated sample and potential environmental source samples. Similarity scores were computed and used to rank potential source samples. The software has been tested on a diverse set of incurred samples by comparing results from the software with those from human experts. We demonstrated that the software provides results consistent with human expert observation. This software also provided the advantage of reliably evaluating an increased sample lot which increased overall efficiency.

## 1. Introduction

Environment pollution has become a significant food safety concern due to pollutants detected in the environment being transferred to the food supply. Exposure to environmental pollutants such as persistent organic pollutants (POPs) [1,2,3,4,5,6] may cause various serious health effects including mutagenic, carcinogenic, and cardiovascular disease as well as endocrine disruption [7,8,9,10]. POPs are a class of carbon-based chemicals that may contain chlorine/bromine/fluorine in their structure. This stable chemical structure makes POPs either resistant to environmental degradation or slow to degrade. Thus, humans and animals can be exposed to POPs over long periods of time. To protect public health, many international and national regulations have been issued to reduce the release of POPs into the environment such as the Stockholm Convention [11,12,13,14], the United Nations Economic Commission for Europe (UNECE) POPs Protocol [15], the Regulation for Registration, Evaluation, Authorization and Restriction of Chemicals (REACH, EC no. 1907/2006 as amended) [16], the EU regulation for the Placing of Plant Protection Products on the Market (EC 1107/2009) [17], and the U.S. Toxic Substances Control Act [18]. As a result of these regulations and legal frameworks, levels of POPs in the environment have been decreasing. However, POPs continue to be of great environmental concern due to their persistence and significant negative impact on human and environmental health.

Human exposure to POPs is mostly through contaminated food, especially fatty foods due to the lipophilic nature of the contaminants. There are many pathways by which POPs can enter the food chain. For example, previously released POPs in the environment can easily be transferred to crops, livestock, and seafood which can eventually pose a significant threat to public health. Aside from the environment, POPs can be introduced to the food supply by human activity. There are many accidents reported such as Yusho disease in 1968 [19] and Yucheng poisoning in 1979 [20] where the public was accidentally exposed to high levels of POPs. Contaminated animal feed supplies can also introduce POPs into food as occurred in the Belgian dioxin crisis in 1999 [21] and the Ireland dioxin accident in 2008 [22]. To ensure food safety, many international and national bodies have set food and animal feed limits for commonly found POPs, and related surveillance programs have been developed to ensure that POPs contamination is below harmful level [23,24,25].

If a food sample is found to contain high concentrations of POPs, it is important to trace sources of contamination so that measures can be taken to remove them and prevent further contamination. Since POPs-contaminated samples exhibit specific congener profiles, the congener-specific concentrations can be used as a fingerprint to trace sources of contamination. For example, the source of dioxin-contaminated milk in the Netherlands was identified to be kaolinic clay used in the bovine feed supply because its congener pattern was similar to the known congener patterns of clays [26]. Recent studies have shown that the congener patterns method is a useful tool to identify sources of contamination [27,28,29,30]. Hoogenboom et al. reviewed some known incidents and showed how visual comparison of congener patterns was used to determine sources of contamination [27]. Besides visual comparison, statistical methods have been used to compare POPs congener patterns including permutational multivariate analysis of variance [31], canonical correlation [32], principal component analysis [28,33,34,35,36], linear discriminant analysis [34,35] and multidimensional scaling [31]. These statistical methods are general tools that have been used to assess the relationship between two datasets. Therefore, some characteristics of POPs are not considered. For example, these methods cannot be used to compare congener patterns from both the concentration and the toxicity level. In addition to this, low level POPs would be ignored in the samples. Therefore, to improve the similarity calculation, we developed a software with algorithms designed specifically for POPs.

In the FDA laboratory, when a food sample is found to contain high levels of POPs, analysts examine the congener pattern of the food sample to infer the source of contamination. The concentration and congener pattern of POPs used in the examination is the excel and python processed targeted mass spectrometry (MS) data [37]. In many cases analysts need to compare the pattern of a contaminated food sample with samples collected by FDA investigators from the same farm or producer. If the pattern of the contaminated sample matches the pattern of a collected sample, the collected sample can be identified as a potential source of contamination. The contaminated sample is a sample that contains elevated naturally incurred levels of some POPs that we would like to find the source. The collected samples are historical samples or newly collected samples by the FDA that can be compared to “Contaminated Sample” in order to determine source of incurred levels in the “Contaminated Sample”. However, this visual comparison is dependent on human expertise. As the size and complexity of collected samples increase, it becomes increasingly difficult to conduct the comparison manually. To overcome this difficulty, we developed software to facilitate the identification of POPs-contaminated sources by computing similarities between a contaminated sample and potential source samples. Different similarity scores were computed and used to rank potential source samples. A similarity rank of potential source samples was provided to analysts for further investigation. Our results demonstrate that this software increased efficiency for the identification of potential sources of POPs contamination.

## 2. Results

### 2.1. Software Details and Interface

The software was developed in Python 3.6 (Python Software Foundation, Beaverton, OR, USA, http://python.org), and the graphical user interface was built using Tkinter module. The source code and Windows executable files are available upon request.

An illustration of the software interface, shown in Figure 1, contains four sections; (1) Data Input, (2) Examination by Visualization, (3) Method for Identification and (4) Run the Software. The first section, “Data Input”, prompts the user to select a contaminated sample of interest, “Enter Target Data File”, while the second prompt “Enter Source Data Files” contains the historical contaminated samples that are used for comparison (referred to as potential source samples hereafter). The sample files are csv files containing three columns: congener names, their concentrations and TEQs. The user can also manually weight results by selecting the “Set Weight” button and choosing congeners of interest. The default weight for each congener is 1, meaning that all congeners are considered equally. If there is a need or desire to focus on specific congeners, this can be accomplished by setting the weights of those congeners to 1 and the other values to zero. After selecting “Next” the user can examine data graphically in the second section, “Examination by Visualization”, by either concentration or by Toxic Equivalent (TEQ). All data are plotted for each congener for all samples provided in the first section. The third section, “Methods for Identification”, allows users to choose ranking methods. Based on the similarity scores computed, there are four ranking methods: (1) TEQ Ranking is based on the TEQ similarity score, (2) Concentration Ranking is based on the concentration similarity score, (3) Both TEQ and Concentration Ranking rank potential source samples equally by both TEQ and concentration similarity scores, (4) Weight-based TEQ and Concentration Ranking apply weights to both TEQ and concentration similarity scores to generate a combined similarity score for ranking samples. The “Number of Samples” ranked can be chosen by the user; however, the default value is set to 5 ranked samples. In the fourth section, “Run” can be selected to view similarity rankings and then saved “Save as…” to a user specified location, or the program can be closed by selecting the “Exit” button.

### 2.2. Performance Test

The software performance was evaluated by comparing results derived from human analysts to those obtained from the software. Rankings of the potential source samples from the software (some shown below) were compared with rankings made by human experts. Use of the software allows many samples to be quickly evaluated for similarities, thus significantly reducing the evaluation time.

#### 2.2.1. Potential Source Sample Test Case

The software was tested on ten contaminated food samples. Here, animal tissue was used as an example to show software performance. Elevated levels of dioxins/furans were found in an animal tissue [38] and sent to FDA for follow-up studies. To trace the potential sources of the elevated dioxins/furans, FDA investigators collected 11 potential source samples such as wood slivers from a utility pole and soil from the same farm. The lab focused on seven dioxins, ten furans, and three dioxin-like PCBs while evaluating the patterns. Similarity scores comparing the contaminated animal tissue TEQ pattern to patterns of matrices collected at the farm are shown in Table 1. Wood slivers from a utility pole showed the highest similarity score at 0.63, indicating that this sample was a potential source of contamination.

The TEQ pattern for the contaminated animal tissue sample was compared to the 11 FDA collected potential source samples using principal component analysis (PCA). As shown in Figure 2, wood sliver from a utility pole sample (solid red downward triangle) is the closest point to the contaminated animal tissue sample (solid red circle), indicating this sample has the most similar pattern as the contaminated sample, indicating the similarity score from the software could be used to represent the similarity in congers pattern.

#### 2.2.2. Potential Source Sample Test Results

For quality control purposes, duplicate samples are tested monthly to evaluate analytical precision in FDA Dioxin laboratory [37]. If the precision is acceptable, the original sample data should be statistically consistent with duplicate sample data and their similarity score should be relatively high. In this study, data from 19 original and duplicate sample pairs were used to test the performance of the software. The sample details for these 19 pairs are shown in Table 2. Some sample pairs with incomplete analytical results were not included in this study. These samples do not necessarily contain high levels of POPs. The software was not developed to show the similarity of duplicates, but historical duplicates were used here to show the reliability of the software in calculating similarity scores. The 38 potential source samples were ranked using weight-based similarity scores that were calculated with the “Concentration weight” of 0.4 and the “TEQ weight” of 0.6. The value of 0.4 and 0.6 are recommended by human expert based on their experience that TEQ played a more important role when they make the decision. Users can choose any weight they like when using the software. The weight-based ranking methods was chosen so that both concentration and TEQ patterns were considered in the similarity calculation. The matrix of similarity scores between the 38 potential source samples are shown in Figure 3.

The similarity scores between the 19 potential source samples and their corresponding duplicates were high (green cells in Figure 3), indicating that the software reported the potential source samples at values very similar to their duplicates as expected. There are 52% of the duplicate (10 pairs) have their similarity scores equal to 1 while 78% (14 pairs) have scores greater than 0.98. We noticed that some similarity scores from non-paired samples were also high. For example, sample **1** is the duplicate of sample **2**, which is animal feed ferrous sulfate. The similarity score between them is equal to 1 indicates that the experiments are of high quality. Interestingly, both samples **1** and **2** showed very high similarity scores at 1 with samples **17** and **18** which are duplicates of a crab sample, **19** and **20** which are duplicates of a cream substitute sample, **21** and **22** which are duplicates of a milk sample, **25** and **26** which are duplicates of a sweet rolls sample, **29** and **30** which are duplicates of a cabbage sample, **31** and **32** which are duplicates of a chocolate candy bar sample, **33** and **34** which are duplicates of another chocolate candy bar sample, **35** and **36** which are duplicates of a English muffin sample (Figure 3). A closer look at the POPs detected in these samples revealed that all of the samples contained very low POPs concentrations. Comparing these samples with randomly selected 100 blank samples tested in the FDA Dioxin laboratory found that the POPs detected were below the mean plus two times standard deviation of POPs levels of the blank samples (Figure 4), indicating these samples likely contain POPs near or below detection limit values. Human experts concluded that these samples were of minimal concern regarding levels of POPs detected. Again, the manual process and software reported consistently similar results.

On the other hand, the software reported similarity scores 0.8252, 0.8007, and 0.7951 for pairs **3** and **4** which are duplicates of an animal feed finisher ration sample, **15** and **16** which are duplicates of a crab sample, and **23** and **24** which are duplicates of a poultry mixed feed ration sample, respectively (Figure 3), indicating the concentrations of congeners for these three sample pairs were not perfectly consistent. Inspection of the data showed that a few congeners showed different concentrations in these pairs of samples (Figure 4). For example, 0.87 pg/g PCB-126 was determined in one duplicate of poultry mixed feed ration sample **23**; but no PCB-126 was detected in the other duplicate sample 24. Although sample **23** showed a slightly higher PCB-126 concentration than the maximum PCB-126 concentration (0.3317 pg/g) among the 100 selected blank samples, levels of the other 19 congeners were below detection limits (compared to blank samples), thus the human experts concluded that the concentration pattern of samples **23** and **24** are similar. The software reported their similarity score at 0.7951, indicating the two samples exhibit a similar POPs pattern. In summary, the comparison between conclusions drawn from the software reports and human expert evaluation showed no discrepancy for any of the 38 potential source samples tested.

## 3. Discussion

Comparison of congener patterns, acquired from MS data, is used to help identify sources of POPs contamination in food. Manual comparison by subject matter experts is time consuming and dependent on the experience of the analysts. The software developed here has increased sample capability, therefore, efficiency while identifying POPs contamination sources in regulatory science. To improve the efficiency of contamination source identification, we developed a software for automatic comparison of patterns of congener concentration determined using excel and python processed MS data [37]. The enhanced software greatly streamlines the process by reducing the data evaluation time. Our tests confirmed that the software can report results consistent with manual human expert evaluation.

The software was tested for comparing patterns of concentrations of dioxin/furan and dioxin-like PCBs in an FDA Dioxin laboratory. However, the software has no restrictions on the types of POPs or the number of potential source samples that can be processed. Once concentration/TEQ data is provided, the software can compute similarity scores based on common congeners for samples in one single run without user intervention. The different types of similarity scores increase the capability of this software well beyond the manual ability of human analysts. Multiple versions of the same data can be easily viewed which increases the opportunity to identify otherwise missed similarities.

There are some limitations of the software. The software ranks potential samples based on their similarity scores in congener concentrations and TEQs compared to a contaminated food sample. It is possible that the top ranked potential source samples are not always the sole source of contamination; this can be due to changes in the congener patterns as the congeners move through the food chain due to their different absorption, distribution, metabolism, and excretion rates [27]. A possible way to overcome this is to introduce animal bioconcentration factors. The software allows users to manually enter transfer rates as weights for congeners. However, due to the difficulty in obtaining the transfer rate for each food sample and each congener, we did not use this as its default setting in our software.

## 4. Materials and Methods

### 4.1. Study Design

This study aims to automate the process of identifying sources of POPs contamination by calculating the similarities between a contaminated sample and potential source samples. Our software is expected to calculate the similarity score between contaminated samples and potential source samples and suggest potential sources ranked by their similarity scores.

The concentration and calculated TEQ obtained from MS data are used to compute the similarity score between a contaminated sample and each potential source sample. TEQ is the product of concentration and toxic equivalency factor (TEF) [39] and is often used to measure the toxicity of a congener. The concentration/TEQ data provided to the software are compared with the concentration/TEQ data from 100 randomly selected blank samples. When a concentration/TEQ is smaller than the expected detection error determined using the 100 randomly selected blank samples (mean + 2 times the standard deviation), the congener should be considered absent from the potential source samples, thus the concentration/TEQ is set to zero. When a concentration/TEQ is larger than the expected detection error, the expected detection error is subtracted from the concentration/TEQ.

After preprocessing the input data, three steps are used to calculate similarity scores in the software. First, concentrations/toxic equivalency (TEQ) are normalized by their maximum value to show the relative contribution of each congener. Second, the normalized concentrations/TEQ are used to calculate the similarity between a contaminated sample and each potential source sample. Then the computed similarity scores are used to rank the potential source samples.

### 4.2. Similarity Calculation

#### 4.2.1. Normalized Concentration/TEQ

In this step, the concentration/TEQ for each congener is normalized by the maximum value in a sample. The software calculates both concentration and TEQ values to provide the user with three options to evaluate the similarity: concentration-based pattern, TEQ-based pattern, and the combined pattern. This normalization step converts the congener concentration/TEQ to a scale between 0 and 1. The normalized concentration/TEQ is calculated by Equation (1):(1)$$c_{i}^{n}=\;\frac{c_{i}^{o}}{c_{i}^{max}}$$
where $c_{i}^{n}$ is the normalized concentration/TEQ for congener *i*; $c_{i}^{o}$ is the original concentration/TEQ for congener *i;*
$c_{i}^{max}$ is the maximum concentration/TEQ of congeners in the sample.

#### 4.2.2. Calculate Similarity

The normalized concentration/TEQ is used to calculate similarity score. This similarity score calculation is based on three hypotheses. First, the overall similarity of two samples is based on the similarities of individual congeners. Second, congeners with higher concentration weights more than the low-level congeners since the source of high-level congeners are more important than that of low-level congeners. Third, the low-level congeners still contribute to the overall calculation. This would enable the software compare samples containing very low POPs concentrations. The similarity $S_{p}^{c}\;$between a potential source sample to a contaminated sample is calculated by Equation (2):(2)$$S_{p}^{c}=\;\overset{n}{\underset{i=1}{\sum }}\frac{w_{i}(c_{i}^{c}+c_{i}^{p}+0.2566)}{\sum (c_{i}^{c}+c_{i}^{p}+0.2566)}\;\left(\frac{2}{1+\left|c_{i}^{c}-c_{i}^{p}\right|}\;-1\right)$$
where n is the number of congeners in the sample; $w_{i}$ is the weight of congener *i*. The value of $w_{i}$ is manually entered by users. Its default value is 1 meaning that all the congeners would be considered equally. $c_{i}^{c}$ and $c_{i}^{p}$ represent the normalized concentration/TEQ for congener ***i*** in a contaminated sample and a potential source sample, respectively. An adjustment factor of 0.2566 is the mean concentration of congeners from 100 randomly selected blank samples. The similarity score is between 0 and 1. High similarity values indicate a greater match is between a contaminated sample and a potential source sample. Here, we calculated two types of similarity score: the TEQ similarity score based on TEQ of each congener, and the concentration similarity score based on the concentration of each congener.

#### 4.2.3. Ranking Potential Source Samples

The software provides four methods for ranking potential source samples. In the first two methods, the potential source samples are ranked using TEQ similarity scores or concentration similarity scores, respectively. The third method ranks the potential source samples using consensus TEQ and concentration similarity scores. In this method, the potential source samples are first ranked separately using their TEQ and concentration similarities. Then samples common to both lists are selected and their average TEQ and concentration similarity scores are used to rank the potential source samples. For example, to find the top 5 potential source samples using the consensus ranking, the software first finds the two top 5 listings using TEQ and concentration ranking. Then potential source samples common to these two lists are ranked using the average value of TEQ and concentration similarity scores. The fourth method ranks potential source samples using weight-based similarity scores that combine concentration and TEQ similarity scores. The weight-based similarity score is calculated using Equation (3):(3)$$S\;=\;\left(TEQ\;weight\right)\times \;S_{TEQ}+\;\left(Concentration\;weight\right)\times \;S_{conc}$$
where $S_{TEQ}$ is the TEQ similarity score and $S_{conc}$ is the concentration similarity score. If the weight-based method is chosen, the software then asks users to enter TEQ and *concentration weights*. If users want to focus on toxicity level, a larger value should be given to *TEQ weight*. After obtaining user input values, the software computes the weight-based similarity scores and ranks the potential source samples.

## 5. Conclusions

This software has increased the sample capability to compare large numbers of potential source samples using their similarity to a contaminated food sample. Similarity scores are calculated between a contaminated sample and potential source samples. Different types of similarity scores can be computed and used to rank potential source samples based on user needs. We tested the software on a diverse set of samples and compared the results from the software with results from human experts. The tests confirmed that the software yielded results consistent with human expert results. Our study results demonstrated that the software is reliable in ranking samples based on their POPs concentration patterns.

## Figures and Tables

**Figure 1 molecules-26-00685-f001:**
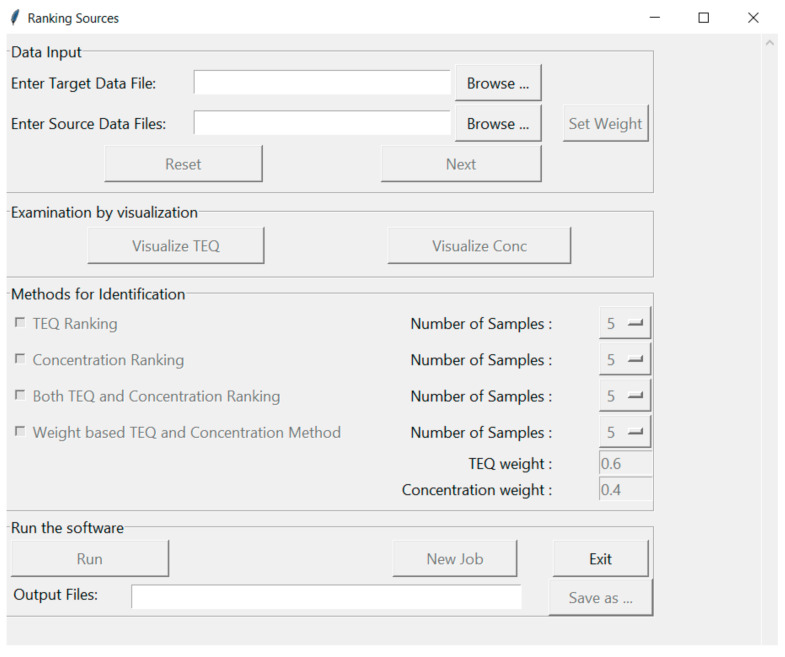
Software interface. Functions are grouped into five sections. Section titles are shown in the top left corners of the section rectangles. Buttons with black text are functions that can be executed. Buttons with grey text are functions that need to be activated after users input data or parameters, or execute functions.

**Figure 2 molecules-26-00685-f002:**
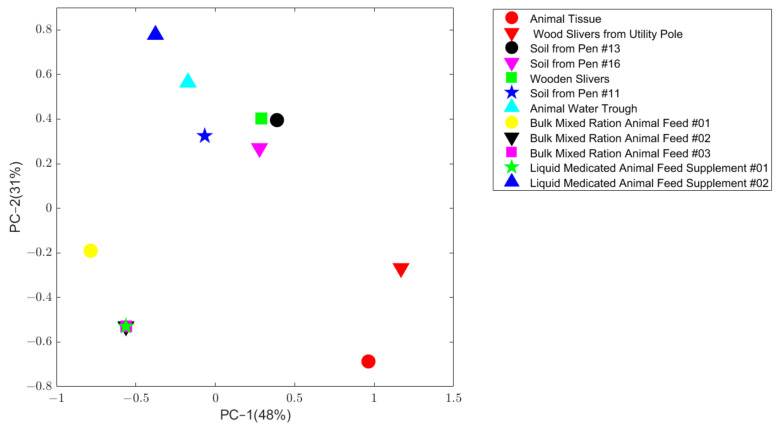
Principle component analysis on the normalized concentration of the animal tissue sample and 11 potential source samples. The x axis is the first principal component, and the y axis is the second principal component. Samples were plotted in different colors and shapes which are given in the legend located at top right.

**Figure 3 molecules-26-00685-f003:**
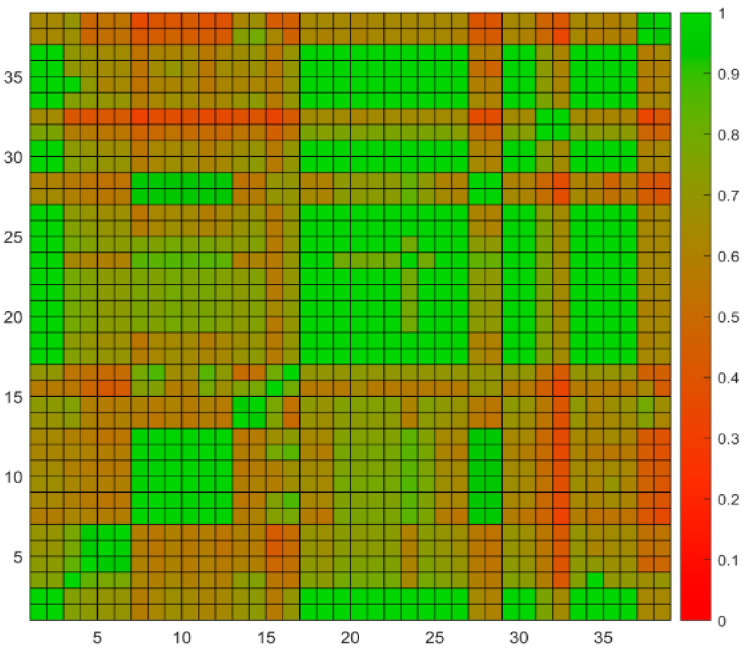
Heatmap of similarity scores. The 19 original potential source samples are shown in odd numbers and their duplicates are shown in even numbers. Similarity scores are coded in colors defined in the right legend.

**Figure 4 molecules-26-00685-f004:**
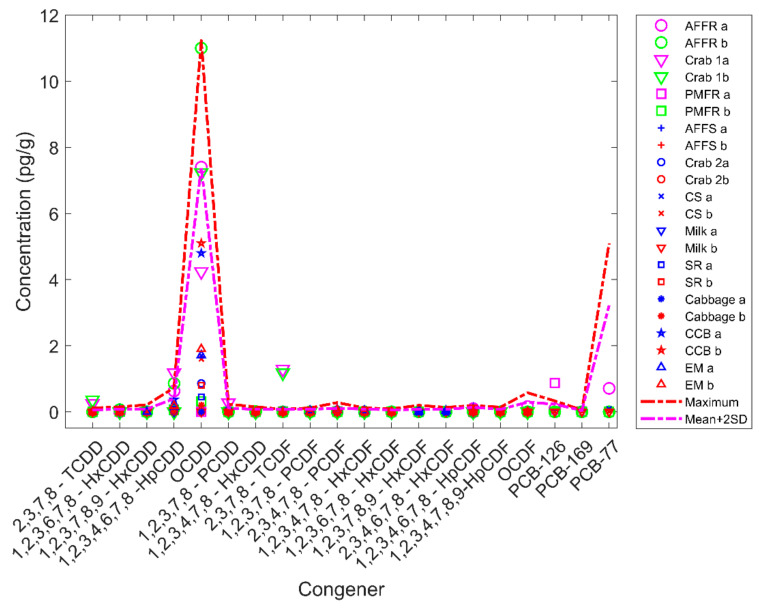
Concentration (pg/g) of congeners acquired from mass spectrometry data. The congeners are labeled on the x-axis. The y-axis shows determined concentrations. Samples are plotted in different shapes with different colors for original and duplicate samples as shown in the right legend panel. The red line shows the maximum concentrations of congeners determined in the 100 randomly selected blank samples. The magenta line depicts mean concentrations plus two times the standard deviation for each of the 100 blank samples. Abbreviations used for sample names: AFFR-animal feed finisher ration; PMFR-poultry mixed feed ration; AFFS-animal feed ferrous sulfate; CS-cream substitute; SR-sweet rolls; CCB-chocolate candy bar; EM-English muffins.

**Table 1 molecules-26-00685-t001:** Ranking and TEQ similarity scores with the animal tissue sample of the 11 potential source samples.

Ranking	Potential Source Sample	Similarity Score
1	Wood Slivers from Utility Pole	0.63
2	Soil from Pen #13	0.58
3	Soil from Pen #16	0.56
4	Wooden Slivers	0.54
5	Animal Water Trough	0.50
6	Bulk Mixed Ration Animal Feed #02	0.50
7	Bulk Mixed Ration Animal Feed #03	0.50
8	Liquid Medicated Animal Feed Supplement #01	0.50
9	Liquid Medicated Animal Feed Supplement #02	0.50
10	Soil from Pen #11	0.49
11	Bulk Mixed Ration Animal Feed #01	0.43

**Table 2 molecules-26-00685-t002:** Samples details for duplicate and original samples.

Sample Number	Sample Matrix
Sample 1	Animal Feed ferrous sulfate duplicate
Sample 2	Animal Feed ferrous sulfate original
Sample 3	Animal feed finisher ration duplicate
Sample 4	Animal feed finisher ration original
Sample 5	Animal feed starter ration duplicate
Sample 6	Animal feed starter ration original
Sample 7	Bluefish 1 duplicate
Sample 8	Bluefish 1 original
Sample 9	Bluefish 2 duplicate
Sample 10	Bluefish 2 original
Sample 11	Bluefish 3 duplicate
Sample 12	Bluefish 3 original
Sample 13	Complete turkey ration duplicate
Sample 14	Complete turkey ration original
Sample 15	Crab 1 duplicate
Sample 16	Crab 1 original
Sample 17	Crab 2 duplicate
Sample 18	Crab 2 original
Sample 19	Cream Substitute duplicate
Sample 20	Cream Substitute original
Sample 21	Milk duplicate
Sample 22	Milk original
Sample 23	Poultry Mixed feed ration duplicate
Sample 24	Poultry Mixed feed ration original
Sample 25	Sweet rolls duplicate
Sample 26	Sweet rolls original
Sample 27	Trout duplicate
Sample 28	Trout original
Sample 29	Cabbage duplicate
Sample 30	Cabbage original
Sample 31	Chocolate candy bar 1 duplicate
Sample 32	Chocolate candy bar 1 original
Sample 33	Chocolate candy bar 2 duplicate
Sample 34	Chocolate candy bar 2 original
Sample 35	English muffins duplicate
Sample 36	English muffins original
Sample 37	Frankfurters duplicate
Sample 38	Frankfurters original

## Data Availability

The data are not publicly available due to data confidentiality requirements.

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
