# Peer review of "Software-Assisted Pattern Recognition of Persistent Organic Pollutants in Contaminated Human and Animal Food"

_molecules, 2021, doi:10.3390/molecules26030685_

Round 1

Reviewer 1 Report

The manuscript is clear and introduces a way to compare profiles of contaminants in different samples, to identify sources of contamination, using a specific software program. There are several studies to compare profiles of contaminants, for instance using PCA, and the authors mention a few of them. The difference between the approach used in other studies in the literature and the approach used in the manuscript should be better discussed. Authors should better explain the novelty of the study: it is for a novel algorithm or is it because the system has been designed specifically for the purpose, while other tools are general ones, and can be used for different purposes, and maybe are less user-friendly, or not supported by collections of data?

Furthermore, authors state that the proposed software can be applied for different POPs. However, their approach refers to TEQ, thus to substances which follow the same mode of action, while POPs may have multiple mode of action. Please clarify this.

Author Response

The manuscript is clear and introduces a way to compare profiles of contaminants in different samples, to identify sources of contamination, using a specific software program. There are several studies to compare profiles of contaminants, for instance using PCA, and the authors mention a few of them. The difference between the approach used in other studies in the literature and the approach used in the manuscript should be better discussed. Authors should better explain the novelty of the study: it is for a novel algorithm or is it because the system has been designed specifically for the purpose, while other tools are general ones, and can be used for different purposes, and maybe are less user-friendly, or not supported by collections of data?

[Author Reply]: Thanks for the suggestion. We added the following discussion in the manuscript’s introduction part.

“Besides visual comparison, statistical methods have been used to compare POPs congener patterns including permutational multivariate analysis of variance[31], canonical correlation[32], principal component analysis[28, 33-36], linear discriminant analysis[34, 35] and multidimensional scaling[31]. These statistical methods are general tools that have been used to assess the relationship between two datasets. Therefore, some characteristics of POPs are not considered. For example, these methods cannot be used to compare congener patterns from both the concentration and the toxicity level. In addition to this, low level POPs would be ignored in the samples. Therefore, to improve the similarity calculation, we developed a software with algorithms designed specifically for POPs.”

Furthermore, authors state that the proposed software can be applied for different POPs. However, their approach refers to TEQ, thus to substances which follow the same mode of action, while POPs may have multiple mode of action. Please clarify this.

[Author Reply]:  Thanks for the reviewer’s comment. We agree with the reviewer that POPs may have multiple mode of action.  In our manuscript, TEQ (Toxic Equivalent) is used to measure the toxicity of a congener based on the 2005 WHO convention. In our software, we provide methods based on TEQ as well as concentrations. For POPs that don’t have TEQ, users can use concentration-based approach to do the comparison.

Reviewer 2 Report

The manuscript describes software to compute a similarity measure between a target sample and potential source samples. The data used in the analysis appears to be from targeted mass spectrometry although this is not mentioned until page 7 of the manuscript and there only referred to as “MS data”. Little detail is given in the Materials and Methods section. Apparently, congener pattern identification is “mostly” done by visual inspection of such data, at least in the FDA laboratory, but, even if not used widely, other relevant methods (e.g. canonical correlation, permutational multivariate analysis of variance found in a quick search) and similarity measures in the literature should be given. Section 2.2.1 presents a sample test case with a table showing calculated similarity to 11 potential sources. The abstract claims that results are consistent with human experts, yet here the only evidence that the scores make sense is provided by PCA. Why is there no comparison with human scoring? If this data (not shown) is anything like that in Figure 4, the PCA will be dominated by OCDD values and the only similarity between the test sample and the wood slivers from the utility pole could be that they have the largest values of this variable – is this enough to identify the source? In the next section a set of samples involving duplicate analyses are used to show that the similarity score for pairs of samples is high in most cases. However, equally high scores are obtained between completely different samples, explained by the fact that the samples are very low in POPs. Why then are these samples included? As the heatmap is only labelled by sample number, a table giving the sample details would be useful here.

When equation (1) is given, it would be useful to state that STEQ and SCONC are to be defined in a later section, otherwise the reader is left wondering what exactly is being combined. There is no explanation for the parameters (0.6 and 0.4) in this equation. How were these chosen? Furthermore, there is no background to the similarity measure give in section 4.2.2. What are the origins of this equation?

The paper is interesting, but requires more explanation on data acquisition and how the similarity measure was developed. The results verified by comparison with human scoring as claimed and other relevant studies should be cited.

Author Response

The manuscript describes software to compute a similarity measure between a target sample and potential source samples. The data used in the analysis appears to be from targeted mass spectrometry although this is not mentioned until page 7 of the manuscript and there only referred to as “MS data”.

[Author Reply]:  Thanks for the reviewer’s comment. The data used in the analysis is excel and python processed targeted mass spectrometry data. We added the following sentence to the manuscript.

“The concentration of POPs used in the examination is the excel and python processed targeted mass spectrometry data.[37]”

Little detail is given in the Materials and Methods section. Apparently, congener pattern identification is “mostly” done by visual inspection of such data, at least in the FDA laboratory, but, even if not used widely, other relevant methods (e.g. canonical correlation, permutational multivariate analysis of variance found in a quick search) and similarity measures in the literature should be given.

[Author Reply]:  Thank for the suggestion. We added the discussion on relevant studies in the manuscript introduction part.

“Besides visual comparison, statistical methods have been used to compare POPs congener patterns including permutational multivariate analysis of variance[31], canonical correlation[32], principal component analysis[28, 33-36], linear discriminant analysis[34, 35] and multidimensional scaling[31]. These statistical methods are general tools that have been used to assess the relationship between two datasets. Therefore, some characteristics of POPs are not considered. For example, these methods cannot be used to compare congener patterns from both the concentration and the toxicity level. In addition to this, low level POPs would be ignored in the samples. Therefore, to improve the similarity calculation, we developed a software with algorithms designed specifically for POPs.”

Section 2.2.1 presents a sample test case with a table showing calculated similarity to 11 potential sources. The abstract claims that results are consistent with human experts, yet here the only evidence that the scores make sense is provided by PCA. Why is there no comparison with human scoring?

[Author Reply]:  Thank the reviewer for the question. We did not compare our results with human scoring because human experts did not calculate scores for the potential sources. They identified the possible source of contamination by visually comparing the contaminated sample with 11 potential sources. Their decision on the possible sources are consistent with our top ranked results. Therefore, in the abstract, we claim our results are consistent with human experts. As inferred by the reviewer, this software allows a concrete, consistent evaluation of comparison, not slighted by analyst bias.  

If this data (not shown) is anything like that in Figure 4, the PCA will be dominated by OCDD values and the only similarity between the test sample and the wood slivers from the utility pole could be that they have the largest values of this variable – is this enough to identify the source?

[Author Reply]:  Thanks for the question. First, the contaminated sample data is not like the data in Figure 4. The data in Figure 4 is duplicates samples data that do not contain high levels of POPs, while the sample case data is contaminated sample data that has high concentrations of 1,2,3,4,6,7,8 – HPCDF(21.839 pg/g), 1,2,3,4,6,7,8 -HPCDD (16.498 pg/g), OCDD(11.867 pg/g) and etc. Second, in our software, the similarity is calculated considering all the POPs in the sample no matter how low the congener concentration is.  Therefore, even for samples dominated by OCDD, the software is able to find samples with similar patterns. Remember also that an OCDD concentration of 11.8pg/g has a TEQ of 0.018pg/g.  If the same sample has a TCDD concentration of 0.08 pg/g (nearly 150 times lower than OCDD) the TCDD TEQ will be 0.08 (8 times greater than OCDD).  Thus, this shows the significance of using weighted evaluations.   (This also addresses reviewer's comment about 0.6 and 0.4 weights)

In the next section a set of samples involving duplicate analyses are used to show that the similarity score for pairs of samples is high in most cases. However, equally high scores are obtained between completely different samples, explained by the fact that the samples are very low in POPs. Why then are these samples included? As the heatmap is only labelled by sample number, a table giving the sample details would be useful here.

[Author Reply]:   Thanks for the reviewer’s question. These duplicate samples with low levels of POPs are real samples that were tested in the lab. We used these samples to show our software is reliable in calculating similarity scores despite the levels of POPs.

Thanks for the reviewer’s suggestion on the sample table. We added a table (Table 2) to provide sample details.

When equation (1) is given, it would be useful to state that STEQ and SCONC are to be defined in a later section, otherwise the reader is left wondering what exactly is being combined. There is no explanation for the parameters (0.6 and 0.4) in this equation. How were these chosen?

[Author Reply]:   Thanks for the reviewer’s comments. To make it clear, we moved the Equation (1) to section 4.2.3 Line 317 and relabeled as Equation 3. The following explanations of STEQ and SCONC  are also in the manuscript. “Where S_TEQ is the TEQ similarity score and S_conc is the concentration similarity score.”

Here, 0.4 of concentration weight and 0.6 of TEQ weight are suggested by human expert based on their experience that TEQ played a more important role when they make the decision. User can choose any weight they like when using the software. The following sentences were added to the manuscript to clarify this. “The value of 0.4 and 0.6 are recommended by human expert based on their experience that TEQ played a more important role when they make the decision. Users can choose any weight they like when using the software.”

Furthermore, there is no background to the similarity measure give in section 4.2.2. What are the origins of this equation?

[Author Reply]:    Thank the reviewer for the question. The similarity measure given in section 4.2.2 is developed by us to quantitatively measure similarities between samples. This similarity calculation is based on three hypotheses.  First, the overall similarity of two samples is based on the similarities of individual congeners. Second, congeners with higher concentration weights more than the low-level congeners since the source of high-level congeners are more important than that of low-level congeners. Third, the low-level congeners still contribute to the overall calculation.  This would enable the software compare samples containing very low POPs concentrations.  We also added this discussion in the manuscript to explain how this equation was developed.

The paper is interesting, but requires more explanation on data acquisition and how the similarity measure was developed. The results verified by comparison with human scoring as claimed and other relevant studies should be cited.

[Author Reply]:    Thanks for the reviewer comment.  As mentioned above, we added data acquisition and the development of the equation in the manuscript. The relevant studies are also discussed and cited in the manuscript.
